# Measurements of Chemical Compositions in Corn Stover and Wheat Straw by Near-Infrared Reflectance Spectroscopy

**DOI:** 10.3390/ani11113328

**Published:** 2021-11-22

**Authors:** Tao Guo, Luming Dai, Baipeng Yan, Guisheng Lan, Fadi Li, Fei Li, Faming Pan, Fangbin Wang

**Affiliations:** 1State Key Laboratory of Pastoral Agricultural Ecosystem, Key Laboratory of Grassland Livestock Industry Innovation, Ministry of Agriculture and Rural Affairs, College of Pastoral Agriculture Science and Technology, Lanzhou University, Lanzhou 730020, China; guot2018@lzu.edu.cn (T.G.); dailm20@lzu.edu.cn (L.D.); yanbp16@lzu.edu.cn (B.Y.); langsh16@lzu.edu.cn (G.L.); lifd@lzu.edu.cn (F.L.); 2Institute of Animal & Pasture Science and Green Agriculture, Gansu Academy of Agricultural Science, Lanzhou 730070, China; pan-faming@163.com; 3Gansu Province Animal Husbandry Technology Extension Master Station, Lanzhou 730030, China; gszxqglk@163.com

**Keywords:** near-infrared reflectance spectroscopy, modified partial least squares, corn stover, wheat straw, nutritional value

## Abstract

**Simple Summary:**

Rapid and non-destructive methods play an important role in assessing forage quality. This study is aimed at establishing a calibration model that predicts the moisture, CP, NDF, ADF, and hemicellulose of corn stover and wheat straw by NIRS. In addition, we also intended to compared the predictive accuracy of combined calibration models to the individual models of chemical compositions for corn stover and wheat straw by NIRS. We show that accurately combining calibrated models would be useful for a broad range of end users. Furthermore, the accuracy of the calibration models was improved by increasing the sample numbers (the range of variability) of different straw species.

**Abstract:**

Rapid, non-destructive methods for determining the biochemical composition of straw are crucial in ruminant diets. In this work, ground samples of corn stover (*n* = 156) and wheat straw (*n* = 135) were scanned using near-infrared spectroscopy (instrument NIRS DS2500). Samples were divided into two sets, with one set used for calibration (corn stover, *n* = 126; wheat straw, *n* = 108) and the remaining set used for validation (corn stover, *n* = 30; wheat straw, *n* = 27). Calibration models were developed utilizing modified partial least squares (MPLS) regression with internal cross validation. Concentrations of moisture, crude protein (CP), and neutral detergent fiber (NDF) were successfully predicted in corn stover, and CP and moisture were in wheat straw, but other nutritional components were not predicted accurately when using single-crop samples. All samples were then combined to form new calibration (*n* = 233) and validation (*n* = 58) sets comprised of both corn stover and wheat straw. For these combined samples, the CP, NDF, and ADF were predicted successfully; the coefficients of determination for calibration (RSQ_C_) were 0.9625, 0.8349, and 0.8745, with ratios of prediction to deviation (RPD) of 6.872, 2.210, and 2.751, respectively. The acid detergent lignin (ADL) and moisture were classified as moderately useful, with RSQ_C_ values of 0.7939 (RPD = 2.259) and 0.8342 (RPD = 1.868), respectively. Although the prediction of hemicellulose was only useful for screening purposes (RSQ_C_ = 0.4388, RPD = 1.085), it was concluded that NIRS is a suitable technique to rapidly evaluate the nutritional value of forage crops.

## 1. Introduction

Cereal crops (namely corn and wheat) are major crops in China, with large amounts of these forage straw materials serving as important roughage sources for ruminant production. About 600 million tons of straw are produced every year in China [1]. However, there are also other uses of cereal straw, such as replacing fossil fuels in the energy sector and chemical industry, and bedding [2]. He et al. demonstrated that basalt fiber addition is an effective way to enhance biohydrogen production from corn straw [3]. H_2_-nanobubble water addition can destroy the cellulose structure of corn straw, reduce the crystallinity of cellulose, and promote hydrolysis [4]. The presence of beneficiary phytochemicals in straw, such as NDF, are important to stimulate rumen fermentation in ruminants. Therefore, an evaluation is required to understand straw sources and purposes [5]. Eastridge et al. reported that feeding different forages (corn silage, alfalfa hay, wheat straw, and corn stover) can result in similar animal performance and ruminal fermentation with adequate formulations of dietary non-fiber carbohydrates and physically effective neutral detergent fiber for dairy cows [6].

Therefore, it is necessary to analyze the nutrient composition of corn stover and wheat straw materials in a timely and accurate manner. The composition of roughage usually varies by harvest time, storage condition, and processing method [7]. However, the analysis of straw chemical composition is time-consuming and costly by conventional analytical techniques, especially when a large number of samples are required. The near-infrared reflectance spectroscopy (NIRS) method is rapid (1 to 2 min per test), non-destructive, low-cost, and in real time [8]. In 1970s, NIRS technology was adopted for the analysis of forages [9]. More importantly, the rapid determination of nutritional compositions of roughage or total mixed ration (TMR) could support accurate nutrition for animal production [10]. The NIRS process relates interactions between diffuse light reflectance in the near-infrared region (750–2500 nm) and biochemical molecules in the forage [11]. Various nutritional components have been estimated in forage by using NIRS, including cellulose, hemicellulose, and ADL in rice straw [12], alfalfa (*Medicago sativa*) [13], and tall fescue (*Festuca arundinacea*) [9]. Research has also shown the potential of using NIRS to evaluate the nutritive value (moisture, ash, ADL, and hemicellulose) of wheat straw, rice straw, and barley straw [2,14,15]. Yet, to our knowledge, there are limited NIRS prediction models for the analysis of corn stover and wheat straw.

The chemical composition of three or more plant materials combined might be predicted by NIRS [16]. Nie et al. improved the NIRS prediction statistics (RSQ_C_, RPD, and slope) by combining timothy and alfalfa in the sample sets, whereas many nutritional fractions were not successfully predicted by separate NIRS equations for each species [17]. Starks and Brown noted that the combined model offered an advantage in improving prediction accuracy for N concentrations of three special cultivars using hyperspectral reflectance from 350 to 1125 nm [18]. Corn stover and wheat straw are likely to have similar biochemical compositions (nutritional value), resulting in similar spectral signatures [19]. It is possible that the combined calibration models of corn stover and wheat straw is better than each respective single-material calibration model.

Therefore, the objective of the present study was to develop a calibration model to use NIRS to predict the moisture, CP, NDF, ADF, and hemicellulose of corn stover and wheat straw collected across China. In addition, this study compared the predictive accuracy of combined calibration models to that of the individual models for determining the chemical composition of corn stover and wheat straw by NIRS.

## 2. Materials and Methods

### 2.1. Sample Collection and Preparation

Wheat straw (*n* = 135) and corn stover (*n* = 156) samples were collected in 2017 at 13 different sites within the provinces / autonomous regions of China, shown in Table 1. The straw materials were cut into 3–5 cm segments and milled through a grinder (CM100, Crinoer technology, Beijing, China) fit with a 1 mm screen prior to nutrient analysis and scanning by a near-infrared spectrometer.

### 2.2. Analyses of Samples by Laboratory Reference Methods

The moisture and CP were determined according to the Association of Official Agricultural Chemists (AOAC) method [20]. Nitrogen was determined using the Kjeldhal method (K9840, Hanon Instrument, Jinan, China) and calculated as CP using the factor 6.25 (AOAC, 1997). The moisture was determined by drying at 105 °C in a forced air oven for 5 h. The NDF and ADF content was measured using a fiber analyzer (A200i, ANKOM Technology, Fairport, NY, USA) according to the methods of Van Soest et al. (2001), which uses sodium sulfite, α-amylase, and sulfuric acid. The ADL content in the ADF residue were determined in accordance with the method described by Ankom Technology [21]. The NDF and ADF were expressed as dry matter percentages to calculate the content of hemicellulose, calculated as follows:Hemicellulose (%) = NDF% − ADF%

### 2.3. Packing and Scanning by Near-Infrared Spectrometer

All straw samples (*n* = 291) were scanned over 850 to 2500 nm at 0.5 nm intervals using a FOSS NIR-Systems DS2500 (FOSS Electric A/S, Hillerød, Denmark). Before sample scanning, the spectrometer was activated, and satisfactory instrument performance was confirmed via instrument response, photometric repeatability, wavelength accuracy tests, and a check-sample scan. Subsequently, the large ring cup (Foss NIR Systems #58374) was overfilled, with an approximately 25 g sample scanned (approximately 10 mm in depth). Triple scans were conducted for each sample, and spectra were averaged before spectra analysis and calibration. Finally, the NIR system referenced the reflectance spectroscopy energy readings for samples to the corresponding readings from the internal standard and recorded the results as the logarithm of the reciprocal of reflectance (log1/R, in which R = reflectance).

### 2.4. Development and Validation of NIRS Calibration Models

The principal component analysis (PCA) scores were calculated by using WinISI IV; software (version4.6.11, Infrasoft International LLC, Silver Spring, MD, USA) for each spectrum. We used PCA for scoring and selecting samples for spectral outliers before calibration and validation. Subsequently, the selected sample sets remaining after the elimination of spectral outliers for moisture, CP, NDF, ADF, ADL, and hemicellulose were sorted and divided into two subsets by reference value: about four-fifths for calibration model development and cross-validation and one-fifth for external validation to test model performance. The corn stover (*n* = 156) and wheat straw (*n* = 135) samples were divided into a calibration set (*n* = 126 of corn stover, *n* = 108 of wheat straw) and a validation set (*n* = 30 of corn stover, *n* = 27 of wheat straw). To improve the accuracy of calibration models, all samples (*n* = 291) were combined to form a new calibration set (*n* = 233) and a new validation set (*n* = 58). Calibration model development for moisture, CP, NDF, ADF, ADL, and hemicellulose used the absorption of diffuse reflection in the near-infrared region (850–2500 nm). A regression method was based on a modification of the partial least squares (MPLS) algorithm, where the spectral data show a higher correction with the reference data and are reduced to variables that account for the main spectral information [22]. In the MPLS regression, the NIR residuals at each wavelength, obtained after each factor were standardized before calculating the next factor (each wavelength was divided by the standard deviations of the residuals). Therefore, MPLS was more stable and accurate than the PLS algorithm [23]. It was used to develop calibration models with the full spectrum for chemical components [17]. To account for possible affecting factors (noise and temperature), a repeatability spectrum was created by collecting three spectra per straw sample. A total of 30 spectral pretreatments were tested to improve the calibration models. The application of detrending to raw spectral data reduces spectral differences related to physical characteristics such as particle size and environment noise. The calibration models were optimized with different scattering correction, mathematical treatment, and regression methods [12]. For scatter correction, we used pretreatments of derivatives and detrending to optimize calibration models. To improve the accuracy of calibration models, 30 spectral pretreatments were tested. The application of derivative to raw spectra increases the complexity of spectra and creates a clear separation between peaks, which overlapped in the information of raw spectra [24]. Mathematical treatments are hereafter referred to using numerals, such as 1, 4, 4, 1, in which the numerals represent the number of the derivative, the gap over which the derivative is calculated, the number of data points in a running average or smoothing, and the number of secondary smoothing points, respectively [23].

The calibration models of moisture, CP, NDF, ADF, ADL, and hemicellulose were considered when it had a lower standard error of calibration (SEC) and the standard error of cross-validation (SECV), a higher coefficient of determination for calibration (RSQ_C_), and a higher value of 1 minus the variance ratio (hereafter referred to as 1-VR). Predicted results were compared with the corresponding reference values as described below. The composition outlier samples were removed from the calibration set if the difference between predicted and reference values exceeded 3 times the SECV, in which case it was removed from the calibration as compositional outlier samples. The following ratio of prediction to deviation (RPD) was utilized to evaluate model quality. Malley et al. suggested this guideline for describing the performance of calibrations for environmental samples: calibrations were excellent when RSQ_C_ > 0.95 and RPD > 4.0, they were successful when RSQ_C_ = 0.90–0.95 and RPD = 3.0–4.0, they were moderately successful when RSQ_C_ = 0.80–0.90 and RPD = 2.25–3.0, and they were moderately useful when RSQ_C_ = 0.70–0.80 and RPD = 1.75–2.25. Some calibrations were only useful for screening purposes, i.e., when RSQ_C_ < 0.70 and RPD < 1.75 [25]. The RPD value was calculated as follows:

RPD = SD/SEP (where SD is standard deviation of the validation sample set) [17].

## 3. Results and Discussion

### 3.1. Laboratory Reference Data

The descriptive statistics including the minimum, maximum, mean, SD, and CV for reference chemical of the calibration set and validation set are shown in Table 2. All component concentrations had wide ranges based on the laboratory analysis. This provided sufficient range to construct prediction models between spectral data and laboratory analyses for each nutrition component [26]. Typically, the mean concentrations of nutritional components were greater in wheat straw than in corn stover, except for CP and moisture, which were lower in wheat straw (3.36% vs. 5.18% and 4.62% vs. 5.35%, respectively) (Table 1). The range of values for the two straw materials were similar, except for NDF, which had a wider range in corn stover (43.73% to 80.71%) than in wheat straw (64.64% to 87.81%). The lower standard deviation values of nutritional components in wheat straw confirmed they varied less than in corn stover, except for ADF and hemicellulose (standard deviation was 46.79% in wheat straw and 36.28% in corn stover).

### 3.2. Spectroscopic Analysis

The average raw reflectance spectrum recorded on air-dried samples is presented in Figure 1. The peaks and valleys present in the spectra demonstrated the different chemical component characteristics of corn stover and wheat straw samples. Typically, near-infrared spectra data can be represented as a function of the wavelength (nm) of diffuse reflection. In the wavelength region 850–2500 nm, there were five main absorption peaks, which were located at wavelengths of approximately 1450, 1900, 2100, 2300, and 2500 nm, respectively. The most groups are O-H (water and carbohydrates), N-H (crude protein), and C-H (ether extract) bands [27].

### 3.3. Development of Calibration Models for Two Straw Materials

The descriptive statistics of the sample number, mathematical treatment, spectrum pretreatment, RSQ_C_, SEC, SECV, and 1-VR for the optimal calibration models are presented in Table 3. As expected, the calibration and cross-validation statistics were different for each component. The first and second derivatives of the spectral data yielded better results than the original spectra for nearly all nutritional components. Generally, the calibration equations from the first or second derivatives generated higher RSQ_C_ and 1-VR and lower SEC and SECV values, especially for the mathematical treatments 1, 4, 4, 1 and 2, 4, 4, 1 [28].

For the moisture of corn stover, the best calibration equation was developed when the mathematical treatment was 2, 4, 4, 1 and when there was no spectrum treatment. However, the mathematical and spectrum treatment settings were different for the moisture of wheat straw (1, 4, 4, 1 and Detrend only, respectively, in Table 2). For the CP, NDF, and ADF of the corn stover, the first derivative yielded the highest RSQ_C_ (0.9572, 0.7861, and 0.8701) and 1-VR (0.9332, 0.7645, and 0.8092), respectively. For the ADL and hemicellulose of corn stover, the best calibration equations were developed when the mathematical treatments were 2, 4, 4, 1 and 0, 0, 1, 1. The two nutritional components of the spectrum treatment were scale and liner. For the CP, NDF, hemicellulose, and moisture of wheat straw, the first derivative (1, 4, 4, 1) produced the highest RSQ_C_ (0.9368, 0.4422, 0.1387, and 0.8569) and 1-VR (0.8870, 0.3753, 0.0531, and 0.8177). The second derivative had a larger RSQ_C_ and 1-VR for the ADF and ADL of the wheat straw.

### 3.4. External Validation of the Calibration Models for Two Straw Materials

The external validation process aims to judge the predictability of the NIRS calibration models [22]. The statistics of external validation (Table 4) were evaluated against the equations developed from the best mathematical treatments and spectrum pretreatments. As could be expected from calibration models (Table 3), the results of validation confirmed that the CP content of corn stover and wheat straw could be accurately predicted by NIRS because the RSQ_C_ was higher than 0.90 and the RPD was larger than 3.0. The CP, as a result of the prediction model, was consistent with previous studies; CP was the best in all models [29]. For NDF and moisture of corn stover, the RSQ_C_ values were 0.7861 and 0.8671, respectively, but the RPD was higher than 2.25. These values confirm that the equations were moderately successful. It was reported that moisture content affects hydrogen content within the sample and that, as a result, band width and position can change [7]. Moreover, the moisture content perhaps changed during sample preparation and scanning from external factors such as temperature, and the humidity of the environment was previously reported to affect the accuracy of moisture determination [30].

A number of researchers have noted that the NDF and ADF of forage could be well predicted by the NIRS technique [11,31,32]. In our study, the equations were moderately useful because the RSQ_C_ values were 0.8701 and 0.5735, but the RPD for the ADF and hemicellulose of the corn stover was higher than 1.75. The equation for ADL was only useful for screening purposes. A poor relationship was similarly noted when silage and barley hay were included within the straw samples [33]. The NIRS method of ADL quantification is related to the spectral changes associated with other components (e.g., CP and NDF) and is thus subject to the accumulated imprecision from multiple components [34]. The literature noted that the 72% sulfuric acid procedure destroys lignin and yields crude lignin, which includes Maillard-type browning products and cutin [35]. Furthermore, the wet chemistry of ADL has low precision, and this polymer is not easily quantified in various types of forages [36]. For wheat straw, the equation could be predicted, but it was less precise than the corn stover for all constituents except for the ADL and moisture models.

### 3.5. Best Calibration Models for Pooled Spectra of Both Corn and Wheat Straw

Dunn et al. and Cozzolino et al. suggested that there was a method for improving calibration model performance by increasing plant species composition [37,38]. The samples from the calibration and validation sets were re-sorted for all straw samples. Corn stover and wheat straw samples were combined for final best calibration model development. The descriptive statistics including the minimum, maximum, mean, SD, and CV of the reference chemicals of the calibration set and the validation set are presented in Table 5.

Compared with the calibration set for each straw material, the range of value for all components, such as moisture, CP, NDF, and ADF, were greatly increased in the combined calibration set. The moisture content ranged from 3.01% to 7.41% in the corn stover and from 2.68% to 7.05% in the wheat straw (Table 2). By combining these two straw materials, the range of values varied from 2.68% to 7.41% (Table 5). The standard deviation values of all components in the combined calibration set were also greater than in the calibration set of each straw.

Based on statistics of NIRS calibration, in which cross-validation (Table 3 and Table 6), and validation (Table 4 and Table 7) used a combined sample set of both corn stover and wheat straw, there were successful equations obtained for CP, ADF, and ADL and moderately useful equations obtained for NDF and moisture, except in regard to hemicellulose. This is because hemicellulose content was obtained by the difference between NDF and ADF, the errors of which accumulated and resulted in poor accuracy for hemicellulose calibration.

For the combined sample set of both corn stover and wheat straw, the content of CP was excellently and successfully predicted by NIRS with a high degree of accuracy (RSQ_C_ = 0.9625 (Table 6), RPD = 6.872 (Table 7)). In the NIR spectrum, the major absorption bands of protein (combination of C–N stretch, N–H in-plane bend, and C–O stretch; combination of C=O stretch and C–H stretch; N–H bend second overtone) could be observed between 2148 and 2200 nm [39,40]. The CP prediction for the combined set was improved compared with the NIRS equation for each straw, which was due to the increased variability (34.19% vs. 23.52% of corn stover, 26.72% of wheat straw) of this component in the combined calibration set [41].

The NIRS prediction of ADF for the combined set of both straw materials outperformed (RSQ_C_ = 0.8745 (Table 6), RPD = 2.751 (Table 7)) the moderately useful NIRS equation for corn stover and the only-useful-for-screening-purposes NIRS equation for wheat straw (Table 3). The ADL NIRS prediction based on the combined species was better (RSQ_C_ = 0.7939 (Table 6), RPD = 2.259 (Table 7)) than the ADL prediction in corn stover and wheat straw (Table 3). The moisture NIRS prediction with the combined set was worse (RSQ_C_ = 0.8342, RPD = 1.868) than those for corn stover (RSQ_C_ = 0.8671, RPD = 2.644) or wheat straw (RSQ_C_ = 0.8569, RPD = 3.024). Although there was an increase in variability (23.60 vs. 21.31 of corn stover and 23.79 of wheat straw) that occurred by expanding the sample numbers, it suggests an overlap of information from the spectroscopy. This result is similar to the outcome from the NDF NIRS prediction.

In the present study, the worst NIRS calibration equation using the combined set was obtained for hemicellulose, with an RSQ_C_ of 0.4388 (Table 6) and an RPD of 1.085 (Table 7). It was determined that the accuracy needed to be improved. The RPD value of hemicellulose was lower than the previous results because the concentration of hemicellulose was calculated as the difference between NDF and ADF from all straw materials [12].

Scatter plots of laboratory reference values versus predicted values of calibration models for a selected set of six nutritional constituents (CP, NDF, ADF, ADL, and hemicellulose) are presented in Figure 2. These results of external validation indicate that the slope regression of the measured vs. predicted values for three nutritional constituents (CP, NDF, and ADF) are close to 1.00 (0.8506–0.9476); the exceptions were ADL (0.7422), hemicellulose (0.2417), and moisture (0.7804). Given the observations of 1-VR, SEC, and SECV, the literature suggests they should be considered with more emphasis and be based on external validation statistics to evaluate the performance of calibration models [23].

## 4. Conclusions

In summary, the descriptive statistics obtained for calibration, cross-validation, and external validation in this research demonstrated that NIRS has a potential for a rapid assessment of forage quality (moisture, CP, NDF, ADF, ADL, and hemicellulose) in corn stover and wheat straw. For corn stover, most of the calibration models had prediction abilities with acceptable accuracy; four of them (moisture, CP, NDF, and ADF) were suitable for quantitative prediction, and the other two (ADL and hemicellulose) were useful for screening purposes. For wheat straw, two of six calibration models (moisture and CP) were adaptable to quantitative prediction, and the other four (NDF, ADF, ADL, and hemicellulose) were useful for screening purposes.

To our knowledge, this is original research aiming to develop NIRS calibration models that might be used to rapidly analyze the nutritional content of corn stover and wheat straw for use as forages. The application of accurate calibrated models combining these straw materials would be greatly useful for a broad range of end users. Increasing sample numbers (variability) by using different straw species improved calibration accuracy. Additionally, it is good for developing precision livestock farming by predicting the components of straw.

## Figures and Tables

**Figure 1 animals-11-03328-f001:**
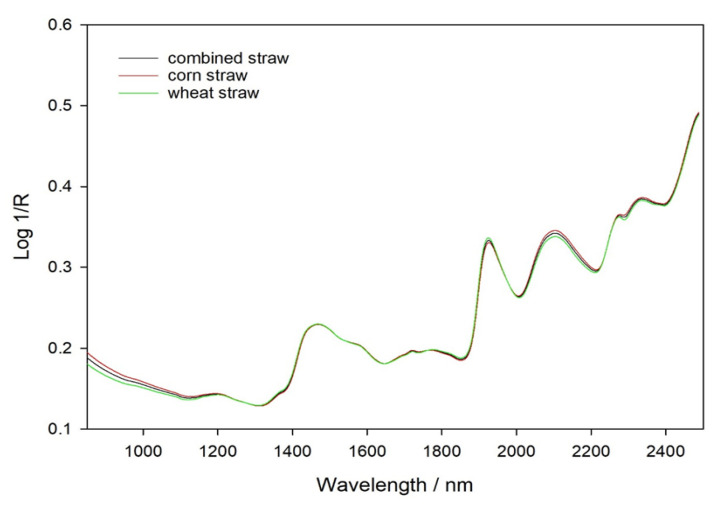
NIRS absorption average spectra of corn stover, wheat straw, and combined straw samples.

**Figure 2 animals-11-03328-f002:**
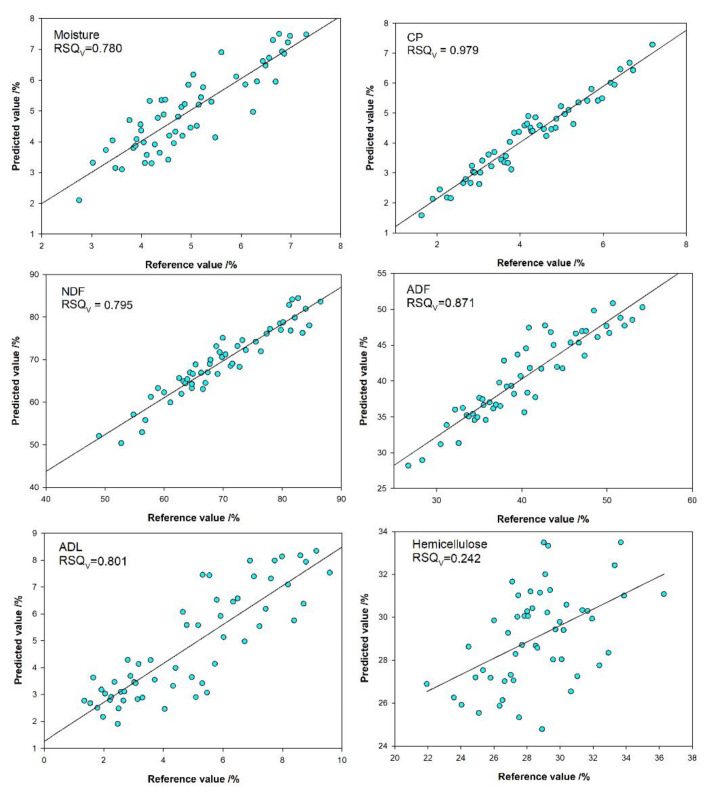
Scatter plot of laboratory reference values versus NIRS predicted values for calibration sets of nutrition quality constituents of all straw samples.

**Table 1 animals-11-03328-t001:** The geographic information of the provinces/autonomous regions from which samples were collected.

Provinces/Autonomous Region	Geographic Information
Gansu	32°31′ to 42°57′ N and 92°13′ to 108°46′ E
Henan	31°23′ to 36°22′ N and 110°21′ to 116°39′ E
Ningxia	35°14′ to 39°23′ N and 104°17′ to 107°39′ E
Shanxi	33°42′ to 34°45′ N and 107°40′ to 109°49′ E
Xinjiang	34°25′ to 48°10′ N and 73°40′ to 96°18′ E

**Table 2 animals-11-03328-t002:** Descriptive statistics for the six constituents of straw used for the development of NIRS calibration and validation models (wet-chemical analysis, DM basis).

Items	Species	Calibration Set						Validation Set					
		*n*	Min (%)	Max (%)	Mean (%)	SD	CV (%)	*n*	Min (%)	Max (%)	Mean (%)	SD	CV (%)
Moisture	Corn stover	121	3.01	7.41	5.35	1.14	21.31	31	3.12	7.31	5.36	1.15	21.43
	Wheat straw	105	2.68	7.05	4.62	1.10	23.79	23	2.75	6.94	4.52	1.14	25.19
CP	Corn stover	123	2.15	10.15	5.18	1.34	25.88	28	2.63	7.19	4.91	1.16	23.52
	Wheat straw	105	1.52	6.75	3.36	0.94	28.14	26	1.62	5.11	3.27	0.87	26.72
NDF	Corn stover	122	43.73	80.71	63.97	6.21	9.70	25	48.93	70.36	62.29	5.51	8.84
	Wheat straw	105	64.64	87.81	77.27	5.94	7.69	21	67.86	86.83	78.52	5.02	6.40
ADF	Corn stover	122	23.36	66.57	36.28	4.71	12.99	29	26.69	42.54	35.54	3.71	10.45
	Wheat straw	105	35.73	58.72	46.79	4.98	10.64	20	39.49	56.78	48.22	4.61	9.56
ADL	Corn stover	121	1.17	10.70	3.26	1.61	49.46	29	1.35	5.79	2.94	1.12	38.12
	Wheat straw	105	4.34	9.93	6.92	1.55	22.34	26	4.40	9.59	6.93	1.56	22.47
Hemicellulose	Corn stover	122	13.53	37.47	27.78	3.58	12.90	26	16.26	30.55	27.29	2.87	10.51
	Wheat straw	105	23.34	44.91	30.58	3.67	12.00	24	25.81	36.26	30.28	2.94	9.70

*n*, number of samples of calibration set; Min, minimum; Max, maximum; SD, standard deviation; CV, coefficient of variation.

**Table 3 animals-11-03328-t003:** Optimal NIRS prediction models of corn stover and wheat straw.

Items	Species	*n*	Mathematical Treatment	Spectrum Treatment	RSQ_C_	SEC	SECV	1-VR
Moisture	Corn stover	117	2, 4, 4, 1	none	0.8671	0.4131	0.5019	0.8020
	Wheat straw	98	1, 4, 4, 1	Detrend only	0.8569	0.4075	0.4575	0.8177
CP	Corn stover	117	1, 4, 4, 1	SNV only	0.9572	0.2543	0.3162	0.9332
	Wheat straw	100	1, 4, 4, 1	SNV only	0.9368	0.2368	0.3151	0.8870
NDF	Corn stover	114	1, 4, 4, 1	Weighted MSC	0.7861	2.7075	2.8284	0.7645
	Wheat straw	104	1, 4, 4, 1	Scale and liner	0.4422	4.6916	4.6249	0.3753
ADF	Corn stover	118	1, 4, 4, 1	Detrend only	0.8701	1.3924	1.6805	0.8092
	Wheat straw	103	2, 4, 4, 1	Standard MSC	0.4266	3.7226	3.8597	0.3776
ADL	Corn stover	118	2, 4, 4, 1	Scale and liner	0.7301	0.6784	1.0306	0.3717
	Wheat straw	102	2, 4, 4, 1	none	0.4829	1.0754	1.1456	0.4074
Hemicellulose	Corn stover	110	0, 0, 1, 1	Scale and Quadratic	0.5735	1.6110	1.6434	0.5521
	Wheat straw	101	1, 4, 4, 1	Scale and Quadratic	0.1387	2.7950	2.9161	0.0531

*n*, number of samples in the calibration set; RSQc, coefficient of determination for calibration; SEC, standard error of calibration; SECV, standard error of cross validation; 1-VR, coefficient of determination for the cross-validation.

**Table 4 animals-11-03328-t004:** Monitoring statistics for the NIR spectroscopic prediction equation for six constituents of corn stover and wheat straw.

Constituent	Species	*n*	Bias	SEP	SEP_C_	Slope	RSQ_V_	RPD
Moisture	Corn stover	31	−0.048	0.435	0.439	0.984	0.854	2.644
	Wheat straw	23	0.028	0.377	0.385	0.903	0.896	3.024
CP	Corn stover	28	−0.102	0.342	0.333	1.037	0.918	3.392
	Wheat straw	26	−0.034	0.235	0.237	1.018	0.927	3.702
NDF	Corn stover	25	−0.426	2.103	2.102	0.925	0.860	2.620
	Wheat straw	21	1.275	2.423	2.112	0.931	0.828	2.072
ADF	Corn stover	29	−0.213	1.739	1.756	0.944	0.779	2.133
	Wheat straw	20	0.781	2.772	2.729	1.252	0.677	1.663
ADL	Corn stover	29	−0.566	1.254	1.139	0.471	0.125	0.893
	Wheat straw	26	0.392	1.299	1.263	0.841	0.355	1.201
Hemicellulose	Corn stover	26	−0.519	1.643	1.590	1.073	0.696	1.747
	Wheat straw	24	0.364	2.550	2.578	1.134	0.232	1.153

*n*, number of samples in validation set; SEP, standard error of prediction; SEPc, standard error of prediction for the bias; RSQv, coefficient of determination of validation; RPD, ratio of prediction to deviation.

**Table 5 animals-11-03328-t005:** Descriptive statistics for the six constituents of straw used for the development of NIRS calibration and validation models.

Items	Calibration Set						Validation Set					
	*n*	Min (%)	Max (%)	Mean (%)	SD	CV (%)	*n*	Min (%)	Max (%)	Mean (%)	SD	CV (%)
Moisture	225	2.68	7.41	5.00	1.18	23.60	56	2.75	7.31	4.98	1.16	23.69
CP	225	1.52	10.15	4.30	1.47	34.19	55	1.62	7.19	4.16	1.34	35.34
NDF	223	43.73	87.81	70.20	8.99	12.81	56	48.93	86.47	70.01	9.07	12.84
ADF	224	23.36	66.57	41.10	7.18	17.47	55	26.69	56.85	40.86	7.05	17.57
ADL	227	1.17	10.70	4.93	2.42	49.09	58	1.35	9.59	4.92	2.41	49.19
Hemicellulose	226	13.53	44.91	29.07	3.87	13.31	52	21.93	36.28	28.59	2.84	13.54

*n*, number of samples of calibration set; Min, minimum; Max, maximum; SD, standard deviation; CV, coefficient of variation.

**Table 6 animals-11-03328-t006:** Optimal NIRS prediction models of straw materials.

Items	Sample Number	Mathematical Treatment	Spectrum Treatment	RSQ_C_	SEC	SECV	1-VR
Moisture	219	1, 4, 4, 1	Detrend only	0.8342	0.4759	0.5421	0.7839
CP	210	1, 4, 4, 1	Weighted MSC	0.9625	0.2708	0.3022	0.9530
NDF	215	1, 4, 4, 1	none	0.8349	3.6973	4.1753	0.7884
ADF	216	2, 4, 4, 1	Scale and Quadratic	0.8745	2.4250	2.9351	0.8154
ADL	215	0, 0, 1, 1	Scale and Linear	0.7939	1.0788	1.1377	0.7697
Hemicellulose	206	0, 0, 1, 1	Standard MSC	0.4388	2.2946	2.3247	0.4212

*n*, number of samples in the calibration set; RSQc, the coefficient of determination for calibration; SEC, standard error of calibration; SECV, standard error of cross validation; 1-VR, coefficient of determination for the cross-validation.

**Table 7 animals-11-03328-t007:** Monitoring statistics for the NIR spectroscopic prediction equation for six constituents of straw materials.

Constituent	*n*	Bias	SEP	SEP_C_	Slope	RSQ_V_	RPD
Moisture	56	−0.035	0.621	0.626	0.769	0.780	1.868
CP	55	−0.004	0.195	0.197	1.017	0.979	6.872
NDF	56	0.464	4.104	4.114	0.977	0.795	2.210
ADF	55	−0.042	2.563	2.586	1.092	0.871	2.751
ADL	58	0.079	1.067	1.074	1.018	0.801	2.259
Hemicellulose	52	−0.484	2.618	2.598	0.634	0.242	1.085

*n*, number of samples in validation set; SEP, standard error of prediction; SEPc, standard error of prediction for the bias; RSQv, coefficient of determination of validation; RPD, ratio of prediction to deviation.

## Data Availability

Not applicable.

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
