# Peer review of "Measurements of Chemical Compositions in Corn Stover and Wheat Straw by Near-Infrared Reflectance Spectroscopy"

_animals, 2021, doi:10.3390/ani11113328_

Round 1

Reviewer 1 Report

In this study the authors use MPLS to calibrate (and validate) models to predict the forage quality of wheat and corn straw (and the combination of both) using NIRS. The in total ~300 analysed samples were collected at 17 sites from different regions within China. The authors were able to successfully predict Crude Protein content for all the different tested materials. The model performance for other biochemical molecules important for forage quality differed depending on if corn or wheat was tested. Yet, in general model performance increased if both corn and wheat samples were combined.

I think the results of this study add more evidence to the ability of NIRS in predicting forage quality. Though, the results are not novel, I want to highlight they are still important. The study could improve by underpinning this importance via highlighting the dichotomy between food supply and bioenergy and how the use of agricultural “byproducts” can be a way forward.

Please see below for section comments and in line comments.

Introduction:

The introduction is in general well written and follows a logical structure, guiding the reader to the aims of this study. Though, I would recommend to strengthen the research need in the first paragraph. The efficient use of agricultural “byproducts” produced along food production is an important topic, especially in the dichotomy between food supply and bioenergy, respectively biofuels.

Materials & methods:

I am wondering if a grinded sample size of 1mm is sufficient for NIRS. From my personal experience with NIRS, there always has been a difficulty with shading within the analysed sample – sort of by “micro-topography”. Additionally, an “overfilled” large ring cup sounds like a cone of sample material, further increasing methodological difficutlies. Though, I am not familiar with the instrument and its technical details used by the authors of this study.

I am missing the information on which criteria the spectra were split into calibration and validation set. If just random, that’s also worth to be named.

Please clarify the mathematical treatments introduced in line 139 – 142. If I understand it correct, this are mathematical operations conducted on the spectra for each sample. Especially clarify why they have been applied. To me it’s unclear how a derivative (e.g. 1st derivative equalling the slope of a curve at point X) can have different data intervals. Further its unclear how different data intervals can have different smoothing points. If the resolution of the NIRS-instrument is 0.5nm, then a data interval of 10 will have 10 smoothing points. Reducing the number of smoothing points therefore would reduce the resolution of the spectral instrument.

Can the authors provide the reader with more information about the actual model applied, the MPLS. How the MPLS is modified and how the regression model is based on it? If possible/available a reference would be sufficient here.

I would recommend to join Table 1 and Table 2. For example, by splitting each (n, min, max, […]) into two columns, with one for validation and the second for calibration set values. This would strengthen the comparability of both, and thereby strengthen the authors important point that both sets are similar. Additionally, the caption should include that this numbers are based on the classic, wet-chemical analysis.

Results:

The results are presented clear. The authors interpret the models output careful and honest, which I appreciate a lot. Though, some of the early sentences within the respective results section, in my view, rather belong to material and methods as they provide information about the way the results are calculated and not the actual results.

A critical concern to raise is, that the presented umbers within the text (e.g. L 280: “RPD = 4.35 [Table 8” of Crude Protein] are not matching the numbers in the referenced Table (RPD = 7.54 for Crude protein). I strongly recommend the authors to carefully check the given numbers within the manuscript and the respectively referenced table.

Conclusions:

I would again recommend the authors to strengthen the broader use of the within the manuscript presented results. “The application of accurate calibrated models combining these 333 straw materials would be greatly useful for a broad range of end users.” As the only sentence putting the findings of this study back into scientific and especially societies context, this requires more elaboration.

Line specific comments:

L19-20: Sentence is unclear to me. Or do you mean that model accuracy improves with the number of samples used for calibration? If so, this holds true for all models, which have a robust pattern underneath.

L21: Why is it crucial to rapid and non-destructive determine forage quality?

L47: Unclear to me. Maybe something into this direction: “[…] requires their rapid determination to be able assign the most efficient use of the straw materials”

L60: Please first write the full name before giving the abbreviation in brackets.

L61: Please formulate more accurate. NIRS process is not per se linked to organic matter in the forage. This is why a calibration model is needed and need to be evaluated. Additionally, everything in forage is “organic matter”, thus I would recommend to reformulate it “biochemical molecules crucial for forage quality”.

L87-90: Geographic information and provinces name can be moved a table or a figure. I would appreciate the latter, as it helps the international reader to visualize the geographical (and thus climatic) range samples have been collected from.

L126-131: The check for outliers (please correct “outliner” to “outlier”) happens before the split into calibration and validation set. Please move this section up to L120 to follow a chronologic structure.

L147-150: This is unclear to me. Spectral samples have been removed from the validation set if their SECV was larger then 3? Was the model afterwards trained again? If so this procedure would reduce the variability present in the validation set and therefore indirectly increase accuracy as the “extreme ends” (SECV>3) are deleted – thus a shrinkage towards the mean happens.

L193-197: Please move this to the according section in Material & Methods.

L197-199: I would rather move the link to the model output table to the end of this results section, then to start with it.

L252: This heading I think is misguiding. If I understood correct the authors pooled all spectra from wheat and corn and afterwards assigned from the pool again a calibration and validation set. “Best calibration models for pooled spectra of both corn and wheat” would be clearer, at least in my view.

L254-257: Please move this to the according section in Material & Methods.

L266: The range of which value? It took me some time to figure out its Moisture content. Please be precise in the formulations used within the manuscript.

Author Response

Manuscript ID: Animals-1457457

Dear editor,

        We thank the referees for their careful reading of our paper. We have carefully considered the comments and have revised the manuscript accordingly. Please finding below our responses to the reviewers’ comments. All the revisions have been addressed in the reply and highlight in the manuscript with yellow background. We hope the revised manuscript can be considered acceptable.

Reply to the comments of Reviewer 1

In this study the authors use MPLS to calibrate (and validate) models to predict the forage quality of wheat and corn straw (and the combination of both) using NIRS. The in total ~300 analysed samples were collected at 13 sites from different regions within China. The authors were able to successfully predict Crude Protein content for all the different tested materials. The model performance for other biochemical molecules important for forage quality differed depending on if corn or wheat was tested. Yet, in general model performance increased if both corn and wheat samples were combined.

I think the results of this study add more evidence to the ability of NIRS in predicting forage quality. Though, the results are not novel, I want to highlight they are still important. The study could improve by underpinning this importance via highlighting the dichotomy between food supply and bioenergy and how the use of agricultural “byproducts” can be a way forward.

Please see below for section comments and in line comments.

Point 1: Introduction:

The introduction is in general well written and follows a logical structure, guiding the reader to the aims of this study. Though, I would recommend to strengthen the research need in the first paragraph. The efficient use of agricultural “byproducts” produced along food production is an important topic, especially in the dichotomy between food supply and bioenergy, respectively biofuels.

Response: Thank you for raising this useful point. According to your suggestion, we modified the introduction in L46-48 (Reference 3 and 4). You can see it below:He et al. demonstrated that basalt fiber addition is an effective way to enhance bio-hydrogen production from corn straw [3]. H2-nanobubble water addition can destroy the cellulose structure of corn straw, reduce the crystallinity of cellulose, and promote the hydrolysis [4].

Point 2: Materials & methods:

I am wondering if a grinded sample size of 1mm is sufficient for NIRS. From my personal experience with NIRS, there always has been a difficulty with shading within the analysed sample – sort of by “micro-topography”. Additionally, an “overfilled” large ring cup sounds like a cone of sample material, further increasing methodological difficutlies. Though, I am not familiar with the instrument and its technical details used by the authors of this study.

Response: We thank you for your careful reading of our paper and providing us with some keen scientific insight. According to your suggestion, we referred the reference (Shi et, al. Major ergot alkaloids in naturally contaminated cool-season barely grain grown under a cold climate condition in western Canada, explored with near-infrared and fourier transform mid-infrared spectroscopy and Ikoyi, A, Y. et, al. Influence of forage particle size and residual moisture on near infrared reflectance spectroscopy (NIRS) calibration accuracy for macro-mineral determination), which grinded sample size of 1mm is sufficient for NIRS. The instrument (FOSS NIR-Systems DS2500) collects spectrum information by diffuse reflection. The overfilled is aimed to make each sample in the scanning process of compaction by the one researcher.

Point 3: I am missing the information on which criteria the spectra were split into calibration and validation set. If just random, that’s also worth to be named.

Response: Thank you for raising this useful point. According to your suggestion, the selected sample sets remaining after elimination of spectral outliers for moisture, CP, NDF, ADF, ADL, and hemicellulose were sorted and divided into two subsets by reference value: about four-fifths for calibration models development and cross-validation and one-fifth for external validation to test model performance in L125-129.

Point 4: Please clarify the mathematical treatments introduced in line 139 – 142. If I understand it correct, this are mathematical operations conducted on the spectra for each sample. Especially clarify why they have been applied. To me it’s unclear how a derivative (e.g. 1st derivative equalling the slope of a curve at point X) can have different data intervals. Further its unclear how different data intervals can have different smoothing points. If the resolution of the NIRS-instrument is 0.5nm, then a data interval of 10 will have 10 smoothing points. Reducing the number of smoothing points therefore would reduce the resolution of the spectral instrument.

Response: We thank you for your careful reading of our paper and providing us with some keen scientific insight. We wrote the manuscript and compared mathematical treatments with other article (using the same calibrated software), which were list in reference. Generally, the mathematical treatments of 1,4,4,1 and 2,4,4,1 can get the best results. (e.g., Reference 23)

Point 5: Can the authors provide the reader with more information about the actual model applied, the MPLS. How the MPLS is modified and how the regression model is based on it? If possible/available a reference would be sufficient here.

Response: Thanks for your suggestion, we have provided more information about the actual model applied of MPLS in L135-141. A regression method based on a modification of the partial least squares (MPLS) algorithm, where the spectral data show a higher correction with the reference data and are reduced to variables that account for the main spectral information (Reference 22). In the MPLS regression, the NIR residuals at each wavelength, obtained after each factor were standardized before calculating the next factor (each wavelength was divided by the standard deviations of the residuals). Therefore, MPLS was more stable and accurate than the PLS algorithm (Reference 23).

Point 6: I would recommend to join Table 1 and Table 2. For example, by splitting each (n, min, max, […]) into two columns, with one for validation and the second for calibration set values. This would strengthen the comparability of both, and thereby strengthen the authors important point that both sets are similar. Additionally, the caption should include that this numbers are based on the classic, wet-chemical analysis.

Response: Thank you for reminding us this important point. According to your suggestion, we have added the value obtained based on wet-chemical analysis in caption and joined Table 1 and Table 2 (Table 2 in the revised manuscript), Table 5 and Table 6 (Table 5 in the revised manuscript) respectively.

Point 7: Results:

The results are presented clear. The authors interpret the models output careful and honest, which I appreciate a lot. Though, some of the early sentences within the respective results section, in my view, rather belong to material and methods as they provide information about the way the results are calculated and not the actual results.

Response: Thanks for your suggestion, we have revised the manuscript in lines 132-136.

Point 8: A critical concern to raise is, that the presented umbers within the text (e.g. L 280: “RPD = 4.35 [Table 8” of Crude Protein] are not matching the numbers in the referenced Table (RPD = 7.54 for Crude protein). I strongly recommend the authors to carefully check the given numbers within the manuscript and the respectively referenced table.

Response: We thank you for your suggestion. I have carefully checked and revised the RPD value in Table 7 (in the revised manuscript).

Point 9: Conclusions:

I would again recommend the authors to strengthen the broader use of the within the manuscript presented results. “The application of accurate calibrated models combining these 333 straw materials would be greatly useful for a broad range of end users.” As the only sentence putting the findings of this study back into scientific and especially societies context, this requires more elaboration.

Response: Thanks for your suggestion, we have strengthened the scientific significance in the conclusions. You can see it below: Additionally, it is good for developing the precision livestock farming by predicting the component of straw.

Line specific comments:

Point 10: L19-20: Sentence is unclear to me. Or do you mean that model accuracy improves with the number of samples used for calibration? If so, this holds true for all models, which have a robust pattern underneath.

Response: Thanks for your suggestion, we mean that the accuracy of calibration models was improved by increasing the sample numbers and variability of different straw species. It was aimed to expand range, which was better for improving the performance of calibration models.

Point 11: L21: Why is it crucial to rapid and non-destructive determine forage quality?

Response: According to your query, rapid, non-destructive methods for determining the forage composition of straw would be crucial to evaluate feed value. The farm can be more precisely feeding, which avoiding the waste of feed resources. We have modified in the revised manuscript in L21-22. You can see it below:

Rapid, non-destructive methods for determining the biochemical composition of straw would be crucial to apply in ruminants’ diet.

Point 12: L47: Unclear to me. Maybe something into this direction: “[…] requires their rapid determination to be able assign the most efficient use of the straw materials”

Response: Thank you for reminding us this important point. We revised as “The presence of beneficiary phytochemicals in straw, such as NDF and ADF, are important to stimulate rumen fermentation in ruminants. Therefore, it requires evaluation to understand straw source and purpose” in L46-48, in the revised manuscript.

Point 13: L60: Please first write the full name before giving the abbreviation in brackets.

Response: Thank you for reminding us this important point. We have corrected total mixed ration (TMR) in L64, revised manuscript.

Point 14: L61: Please formulate more accurate. NIRS process is not per se linked to organic matter in the forage. This is why a calibration model is needed and need to be evaluated. Additionally, everything in forage is “organic matter”, thus I would recommend to reformulate it “biochemical molecules crucial for forage quality”.

Response: We thank you for your careful reading of our paper and providing us with some keen scientific insight. According to your suggestion, we have revised in L65.

Point 15: L87-90: Geographic information and provinces name can be moved a table or a figure. I would appreciate the latter, as it helps the international reader to visualize the geographical (and thus climatic) range samples have been collected from.

Response: We appreciate to review our manuscript, and positive feedback. We have revised in Table 1.

Point 16: L126-131: The check for outliers (please correct “outliner” to “outlier”) happens before the split into calibration and validation set. Please move this section up to L120 to follow a chronologic structure.

Response: Thank you for reminding us this important point. According to your suggestion, we have corrected the “outliers” and moved this section up to L125.

Point 17: L147-150: This is unclear to me. Spectral samples have been removed from the validation set if their SECV was larger than 3? Was the model afterwards trained again? If so this procedure would reduce the variability present in the validation set and therefore indirectly increase accuracy as the “extreme ends” (SECV>3) are deleted – thus a shrinkage towards the mean happens.

Response: Thank you for bringing these meaningful points to our attention. According to your query, spectral samples will remove from the calibration in the internal cross-validation, if their SECV was larger than 3. We have corrected the L161-164 in the revised manuscript.

Point 18: L193-197: Please move this to the according section in Material & Methods.

Response: Thank you for reminding us this important point. According to your suggestion, we have moved this to the according section in Material & Methods. The sentence was displayed in L143-145.

Point 19: L197-199: I would rather move the link to the model output table to the end of this results section, then to start with it.

Response: Thanks for your suggestion. We have modified in the revised manuscript.

Point 20: L252: This heading I think is misguiding. If I understood correct the authors pooled all spectra from wheat and corn and afterwards assigned from the pool again a calibration and validation set. “Best calibration models for pooled spectra of both corn and wheat” would be clearer, at least in my view.

Response: Thank you for reminding us this important point. According to your suggestion, we have revised as “Best calibration models for pooled spectra of both corn and wheat” in L257.

Point 21: L254-257: Please move this to the according section in Material & Methods.

Response: We thank you for raising this useful point. According to your suggestion, we have moved this in Material & Methods.

Point 22: L266: The range of which value? It took me some time to figure out its Moisture content. Please be precise in the formulations used within the manuscript.

Response: Thank you for bringing these meaningful points to our attention. According to your query, we have corrected in the revised manuscript. The species range value compared with combining range in L265-268.

We have tried our best to carefully address all questions/comments/concerns that the reviewers raised and sincerely hope that revised manuscript is acceptable for publication in Animals.

Best wishes!

Yours sincerely

Tao Guo

Reviewer 2 Report

It's a good idea to measure the corn stover and wheat straw with near infrared spectroscopy. Good scientific work, but there were few questions and comments.

Abstract

L26 RPD abbreviation was not defined in this line. Did RPD mean not ratios of prediction to deviation but the ratio of performance to deviation?

Materials and methods

L96 Is there multicollinearity by defining the amount of hemicellulose from NDF and ADF?

L102-104 What is the filling rate of the sample? Was it possible to fill each sample with the same filling rate of corn stover and wheat straw?

L112-115 What was the criteria for separating calibration set and validation set? How about the ratio 13 different site? Did the validation set include every 13 site?

L131-134 Was the spectrum treatment also mathematical calculation?

What is the reason for separating mathematical treatment and spectrum treatment?

In my opinion, the number of derivative intervals, smoothing point is small compared to the size of the spectrum data. It would be on the order of 10 units.

Results

You should round 2 or 3 digital unit to the significant figures in all table.

In table 1, what was the coefficient of variation?

In table 3, where was the optimal component or factor ?

In Figure 2, what was the R^2? Did it show the value of RSQv? It is better to align the scales of the x-axis and the y-axis.

L253 Section after the combination of corn stover and wheat straw. Was it possible to deepen the discussion by absorbing NIR wavelength between the including chemicals an, loading coefficient or regression coefficient?

Author Response

Manuscript ID: Animals-1457457

Dear editor,

        We thank the referees for their careful reading of our paper. We have carefully considered the comments and have revised the manuscript accordingly. Please finding below our responses to the reviewers’ comments. All the revisions have been addressed in the reply and highlight in the manuscript with yellow background. We hope the revised manuscript can be considered acceptable.

Reply to the comments of Reviewer 2

Comments and Suggestions for Authors

It's a good idea to measure the corn stover and wheat straw with near infrared spectroscopy. Good scientific work, but there were few questions and comments.

Abstract

Point 1: L26 RPD abbreviation was not defined in this line. Did RPD mean not ratios of prediction to deviation but the ratio of performance to deviation?

Response: We thank you for your careful reading of our paper and providing us with some keen scientific insight. According to your suggestion, the RPD abbreviation was defined in line 26. The RPD means ratio of prediction to deviation.

Materials and methods

Point 2: L96 Is there multicollinearity by defining the amount of hemicellulose from NDF and ADF?

Response: Thank you for bringing these meaningful points to our attention. According to your query, NDF includes ADF and hemicellulose, so we calculated as follows: Hemicellulose (%) = NDF (%)-ADF (%).

Point 3: L102-104 What is the filling rate of the sample? Was it possible to fill each sample with the same filling rate of corn stover and wheat straw?

Response: Thank you for bringing these meaningful points to our attention. According to your query, the samples were filled about 1/3-2/3 in the large sample cup (approximately 10 mm in depth) in L114-115(in the revised manuscript). To reduce scanning error, all samples were fitted and collected spectrum by the one researcher.

Point 4: L112-115 What was the criteria for separating calibration set and validation set? How about the ratio 13 different site? Did the validation set include every 13 site?

Response: According to your query, the spectrum was split into calibration and validation by using WinISI â…£ software. The calibration and validation sets were sorted and divided into two subsets by reference value: about four-fifths for calibration models development and cross-validation and one-fifth for external validation to test model performance (L125-129).

Point 5: L131-134 Was the spectrum treatment also mathematical calculation?

Response: According to your query, the spectrum pretreatment and mathematical were calculated together.

Point 6: What is the reason for separating mathematical treatment and spectrum treatment?

Response: Thank you for bringing these meaningful points to our attention. According to your query, the spectrum pretreatment and mathematical were calculated together. A total of 30 pretreatments (10 spectrum treatments and 3 mathematical treatments) were tested to improve the performance of calibration models.

Point 7: In my opinion, the number of derivative intervals, smoothing point is small compared to the size of the spectrum data. It would be on the order of 10 units.

Response: Thank you for bringing these meaningful points to our attention. We wrote the manuscript and compared mathematical treatments with other article (using the same calibrated software), which were list in reference. Generally, the mathematical treatments of 1,4,4,1 and 2,4,4,1 can get the better results (e.g., Reference 23).

Results

Point 8: You should round 2 or 3 digital unit to the significant figures in all table.

Response: We appreciate to review our manuscript, and positive feedback. We have revised in all tables (in the revised manuscript).

Point 9: In table 1, what was the coefficient of variation?

Response: According to your query, the coefficient of variation calculated as follows: CV (%) = ( SD / Mean ) *100%. The CV is variable of a group data, which display the degree of data discrete.

Point 10: In table 3, where was the optimal component or factor?

Response: According to your query, the only optimal treatment was shown in Table 3. A total of 30 treatments’ results were compared and selected the optimal treatment, which according to the 1-VR, SEC, SECV and RSQC. Then, the models were validated by the validation sets.

Point 11: In Figure 2, what was the R^2? Did it show the value of RSQv? It is better to align the scales of the x-axis and the y-axis.

Response: Thank you for reminding us this important point. According to your suggestion, we have carried out the value of RSQV in Figure 2. R2 is the correlation coefficient of two groups of data.

Point 12: L253 Section after the combination of corn stover and wheat straw. Was it possible to deepen the discussion by absorbing NIR wavelength between the including chemicals an, loading coefficient or regression coefficient?

Response: Thank you for bringing these meaningful points to our attention. According to your suggestion, we have revised in L280-283 (in the revised manuscript).

We have tried our best to carefully address all questions/comments/concerns that the reviewers raised and sincerely hope that revised manuscript is acceptable for publication in Animals.

Best wishes!

Yours sincerely

Tao Guo

Round 2

Reviewer 1 Report

Thank you for thoroughly adressing the previous feedback.

The manuscript has improved substantially.